# Path-Tracing Distillation: Enhancing Stability in Text-to-3D Generation by Mitigating Out-of-Distribution Issues

## Abstract

Text-to-3D generation techniques signify a pivotal advancement in creating 3D models from textual descriptions. Contemporary state-of-the-art methods utilize score distillation processes, leveraging 2D priors to generate 3D assets. However, these approaches frequently encounter instability during the initial generation phases, primarily due to an distribution discrepancy between the score prediction network and rendered images. Specifically, the raw rendered images of the initial 3D model lie out of the distribution (OOD) of the pretrained score prediction network, which is trained on high-fidelity realistic images. To address this OOD issue, we introduce an innovative Path-Tracing Distillation (PTD) technique that refines the distillation process. Our method sequentially optimizes the 3D model using intermediate score networks that exhibit closer distributional alignment, thereby accelerating the convergence during the early stages of training. This approach not only ensures a more stable increase in CLIP similarity initially but also preserves the visual quality and diversity of the generated models. Comprehensive experiments demonstrate that PTD significantly enhances both the stability and quality of text-to-3D generation, outperforming existing baselines in CLIP scores.

## 1 Introduction

The emergence of 3D asset generation (Qian et al., 2024) has precipitated transformative shifts within the graphics industry, promising a future where 3D models are increasingly synthesized rather than traditionally rendered. A 3D asset, meticulously detailed with intricate textural properties, serves as a foundational element for a myriad of applications spanning animation, virtual reality, gaming, and more. Among the diverse array of techniques, text-to-3D generation (Wang et al., 2023) stands out as a promising approach, offering an efficient and user-friendly mechanism to create 3D models from textual descriptions. This methodology not only streamlines the creative workflow but also democratizes 3D content creation, making it accessible to a broader creators.

Currently, there are two predominant strategies for generating 3D assets from textual descriptions: directly training a generative model on 3D data and constructing 3D models based on 2D priors. The latter approach leverages large-scale diffusion models such as Stable Diffusion (Rombach et al., 2022b) and Imagen (Saharia et al., 2022), which generate multi-view supervised signals to guide the optimization of differentiably rendered images from an evolving 3D model. As the optimization process iteratively refines the 3D model from various perspectives, it progressively converges toward a realistic and coherent representation. This iterative refinement is crucial for ensuring that the generated 3D models accurately reflect the semantic content of the input text while maintaining high visual fidelity across different viewpoints.

At the forefront of these advancements is DreamFusion (Poole et al., 2022), which introduced Score Distillation Sampling (SDS) to predict scores in noisy images rendered from 3D models. Despite its innovative approach, DreamFusion's technique sometimes results in images exhibiting over-saturation or excessive smoothing, detracting from the model's realism and detail. To address these limitations, subsequent methods such as ProlificDreamer (Wang et al., 2023) introduced Variational Score Distillation (VSD), modeling the distribution of multiple generated 3D representations to enable more diverse and accurate 3D model creation. Additionally, LucidDreamer (Liang et al., 2024) proposed

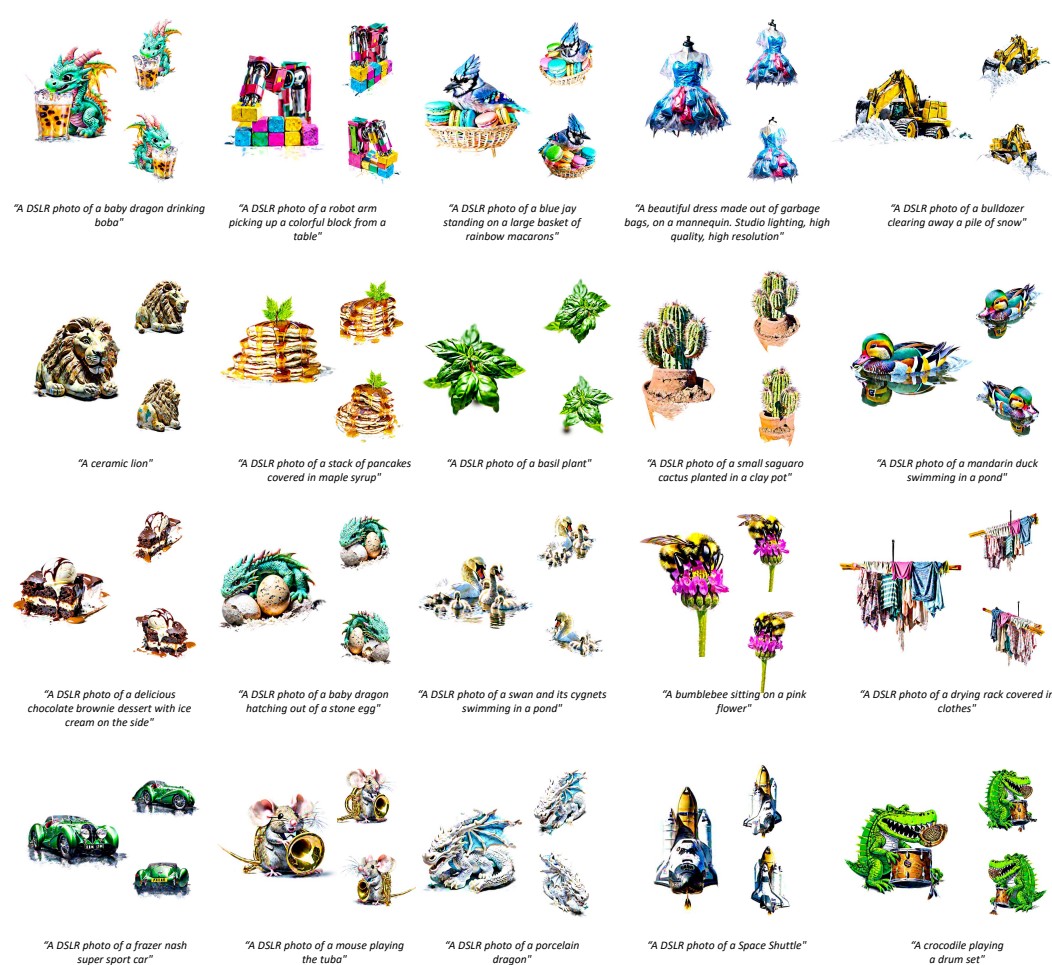

*"A DSLR photo of a baby dragon drinking boba"*    *"A DSLR photo of a robot arm picking up a colorful block from a table"*    *"A DSLR photo of a blue jay standing on a large basket of rainbow macarons"*    *"A beautiful dress made out of garbage bags, on a mannequin. Studio lighting, high quality, high resolution"*    *"A DSLR photo of a bulldozer clearing away a pile of snow"*

*"A ceramic lion"*    *"A DSLR photo of a stack of pancakes covered in maple syrup"*    *"A DSLR photo of a basil plant"*    *"A DSLR photo of a small saguaro cactus planted in a clay pot"*    *"A DSLR photo of a mandarin duck swimming in a pond"*

*"A DSLR photo of a delicious chocolate brownie dessert with ice cream on the side"*    *"A DSLR photo of a baby dragon hatching out of a stone egg"*    *"A DSLR photo of a swan and its cygnets swimming in a pond"*    *"A bumblebee sitting on a pink flower"*    *"A DSLR photo of a drying rack covered in clothes"*

*"A DSLR photo of a frazer nash super sport car"*    *"A DSLR photo of a mouse playing the tuba"*    *"A DSLR photo of a porcelain dragon"*    *"A DSLR photo of a Space Shuttle"*    *"A crocodile playing a drum set"*

Figure 1: Example 3D models generated by our proposed PathTracing Distillation. Rendered image in three views and the corresponding text prompts are presented. Best viewed magnified on screen.

Interval Score Matching (ISM), employing deterministic diffusing trajectories and interval-based score matching to mitigate the over-smoothing effect observed with SDS. More recently, Classifier Score Distillation (CSD) (Yu et al., 2023b) was developed as an implicit classification model for generation, achieving commendable results by refining the score prediction process. However, despite these advancements, existing methods still grapple with instability during the early stages of 3D model generation and require extended periods to achieve stable and high-quality outcomes.

We argue that the score prediction network struggles to accurately predict the noise added to initially rendered images. Such argument is supported by our empirical evidence. Our findings reveal a substantial disparity between the scores predicted by Stable Diffusion and an approximated score (Xu et al., 2023b), given images rendered from the initial raw 3D models. This discrepancy becomes increasingly pronounced when the added noise scale is small. In contrast, for images generated directly by Stable Diffusion, the disparity between predicted and approximated scores is significantly reduced. These observations conclude that the score prediction for images from the initial raw 3D models is inaccurate. An intuitive explanation for this issue is that the rendered images from the low-quality initial 3D model tend to fall outside the support of Stable Diffusion which is trained exclusively on high-quality 2D images.

To address the identified issue, we propose Path-Tracing Distillation (PTD) method, which employs a series of score prediction networks instead of the solitary pretrained score network to perform score distillation. Our core insight is, while the pretrained score network suffers from discrepancy, there

are multiple intermediate distributions between the pretrained score network and the distribution of rendered images. The 3D models may be guided by the intermediate score networks from close to far and finally reach the pretrained score network. This is like when traveling to a remote city without available direct flights, one may transfer several times through intermediate cities along the journey. A planning of travel journey may start from the source to the destination, or from the destination back to the source. We propose to obtain the intermediate score networks in a backward order.

Based on the above insight, we design a two-stage text-to-3D generation pipeline. In the first stage, given a raw 3D model initialized by SF3D (Boss et al., 2024), we finetune the pretrained score model to fit the initially rendered images, thereby gradually degrading the pretrained score network from high-quality real images to low-quality rendered images. Checkpoints are saved during this finetuning, and they are also score networks which altogether represent a transformation path of intermediate distributions connecting the high-quality pretrained score to the low-quality untrained rendered image distribution. In the second stage, the 3D model is iteratively optimized using these checkpoints of score networks as distillation targes in a reverse order one by one. The switch from a current checkpoint to a next one occurs when rendered multi-view images nearly converges to the current checkpoint. The end of the reverse path is the original pretrained score network, so the proposed distillation approach share the same convergence target with other score distillation strategy. This distillation approach encourages a more stable optimization process, effectively mitigating the OOD issue and resulting in high-quality 3D models.

Compared to existing text-to-3D methods, our Path-Tracing Distillation approach predict more accurately score and accerlarate the convergence speed in 3D generation while maintaining both diversity and quality. The proposed approach can be integrated with other existing score distillation 3D generation methods. Our contributions are summarized as follows:

1. We identify the out-of-distribution (OOD) issue during the 3D generation process with empirical evidence.

2. We propose Path-Tracing Distillation (PTD) to mitigate the instability of score prediction in the early stages of generation.

3. Experiments demonstrate that our PTD approach exceeds SOTA methods in CLIP similarity, indicating higher quality of 3D assets.

## 2 PRELIMINARIES

The text-to-3D generation task aims to create a 3D model based on a given text prompt $y$. We denote the parameters of the 3D representation as $\theta$. Our method renders images from specific camera angles $c$ using volumetric rendering, represented as $\boldsymbol{x}_0 := \boldsymbol{g}(\theta, c, y)$. Given the distribution of the training images conditioned on the text prompt, our objective is to minimize the KL divergence between the training image distribution, $p(\boldsymbol{x}_0 \mid y)$, and the rendered image distribution, $q^\theta(\boldsymbol{x}_0)$:

$$\min_\theta \mathrm{KL}\left(q^\theta(\boldsymbol{x}_0) \parallel p(\boldsymbol{x}_0 \mid y)\right) \tag{1}$$

Intuitively, this means that the rendered image $\boldsymbol{x}_0$ should appear realistic from various viewpoints $c$ compared to the training images given the specific prompt.

Advanced techniques typically employ implicit representations like Neural Radiance Fields (NeRF) (Mildenhall et al., 2021) or explicit representations such as 3D Gaussians (Kerbl et al., 2023) to model 3D objects or scenes. To sovle the Eq. (1) and avoid the high dimensional problem, an image $\boldsymbol{x}_0$ undergoes a noise addition process: $\boldsymbol{x}_t = \alpha_t \boldsymbol{x}_0 + \sigma_t \boldsymbol{\epsilon}$, where $\alpha_t$ and $\sigma_t$ follow the diffusion model's schedule (Ho et al., 2020), and $t$ represents the timestep.

DreamFusion (Poole et al., 2022) introduces Score Distillation Sampling (SDS), which leverages a pre-trained model to predict noise $\boldsymbol{\epsilon}_\phi(\boldsymbol{x}_t, t, y)$ on noisy images, guided by the text prompt $y$. SDS calculates gradients by comparing the predicted noise with the actual added noise, updating the 3D representation as follows:

$$\nabla_\theta \mathcal{L}_{\mathrm{SDS}}(\theta) := \mathbb{E}_{t, \boldsymbol{\epsilon}, c}\left[\omega(t)\left(\boldsymbol{\epsilon}_{\mathrm{pretrain}}\left(\boldsymbol{x}_t, t, y\right) - \boldsymbol{\epsilon}\right) \frac{\partial \boldsymbol{g}(\theta, c)}{\partial \theta}\right] \tag{2}$$

ProlificDreamer (Wang et al., 2023) further advances this by proposing Variational Score Distillation (VSD). This method models the distribution of 3D scenes using multiple particles and employs

an auxiliary score prediction network $\epsilon_\phi(\boldsymbol{x}_t, t, c, y)$ to model multiple images rendered from these particles. The auxiliary network is designed as a LoRA on the base network. The optimization of $\epsilon_\phi$ and each particle's parameters $\theta^{(i)}$ is performed alternately, with the gradient of $\theta$ being:

$$\nabla_\theta \mathcal{L}_{\text{VSD}}(\theta) \coloneqq \mathbb{E}_{t,\epsilon,c}\left[\omega(t)\left(\epsilon_{\text{pretrain}}\left(\boldsymbol{x}_t, t, y\right) - \epsilon_\phi\left(\boldsymbol{x}_t, t, c, y\right)\right)\frac{\partial \boldsymbol{g}(\theta, c)}{\partial \theta}\right] \quad (3)$$

LucidDreamer (Liang et al., 2024) proposes Interval Score Matching to minimize the interval score between adjacent timestamps using DDIM (Song et al., 2020) which tackles the over-smooth issue:

$$\nabla_\theta \mathcal{L}_{\text{ISM}}(\theta) \coloneqq \min_{\theta \in \Theta} \mathbb{E}_{t,c}\left[\omega(t)(\epsilon_\phi(\boldsymbol{x}_t, t, y) - \epsilon_\phi(\boldsymbol{x}_s, s, \emptyset))\frac{\partial \boldsymbol{g}(\theta, c)}{\partial \theta}\right]. \quad (4)$$

Based on these problem formulation and advanced distillation techniques, our approach builds a series of score prediction networks to address the out-of-distribution (OOD) issue, bridging the gap between high-quality and low-quality image distributions during the 3D model generation process.

# 3 THE OUT-OF-DISTRIBUTION ISSUE

The out-of-distribution issue often refers to the machine learning model receives data which deviates significantly from the model's training set so the model fails to give reliable predictions. Specifically in our scenario, the OOD issue refers to the score prediction network $s_{\text{pretrain}}$ may receive rendered images different from images in the training set so it gives unstable score. Intuitively, $s_{\text{pretrain}}$ is a large-scale network trained on high-quality realistic images to capture 2D visual prior. Whereas, the initial 3D models, given different representation or initialization, are of low quality. Thus, $s_{\text{pretrain}}$ may not predict scores of the noisy rendered images well enough.

To investigate the OOD issue, we compare the predicted score $s_{\text{pretrain}}(\boldsymbol{x}_t, t, y)$ with the approximated score $s_{\text{approx}}(\boldsymbol{x}_t, t, y)$. The approximated score is computed with Stable Target Field (Xu et al., 2023b) (Appendix G), which includes an additional reference batch of training samples used to calculate weighted conditional scores as the approximation. The comparison is reflected by a proposed matching loss between the predicted and approximated score, formally defined as

$$\mathcal{L}_{\text{matching}}(\boldsymbol{x}_t) = \mathbb{E}_{x_0, t, \epsilon}\|s_{\text{pretrain}}(\boldsymbol{x}_t, t, y) - s_{\text{approx}}(\boldsymbol{x}_t, t, y)\|_2, \quad (5)$$

Here $t$ indicates the timestep used for adding noise. If the noisy image given to the score prediction network lies in the support of the training distribution, the predicted score is expectedly close to the approximated score, and vice versa. The matching loss on the noisy rendered images $\boldsymbol{x}_t^\theta$ from initial 3D models is $3.234 \pm 0.042$. This is significantly higher than the loss value $1.687 \pm 0.075$ on noisy generated images $\boldsymbol{x}_t^{\text{pretrain}}$ given by the pretrained model as a baseline. Detailedly, this difference becomes more pronounced when $t$ is small, and vice versa. This observation suggests the pre-trained model's score prediction accuracy for rendered images is markedly inferior to that for real images. We argue that initially rendered images are out of the distribution of $s_{\text{pretrain}}$ and the score prediction is unstable. Thus, the early generation of 3D models are impeded. To solve this problem, we design the path-tracing strategy as detailed in the next section.

# 4 METHOD

## 4.1 OVERVIEW

We design a two-stage text-to-3D generation pipeline (Figure 2). In the first stage of forming path, given a raw initialized 3D model, we finetune the pretrained score model to fit the initially rendered images and save checkpoints during this finetuning to form the transformation path. In the second stage of tracing path, the 3D model is iteratively optimized using these checkpoints of score networks as distillation targes in a reverse order one by one. The final target is the pretrained score network.

## 4.2 FORMING PATH

The target of this stage is to obtain the transformation path of score networks connecting the pretrained score network $s_{\text{pretrain}}$ and the initial rendered image distribution. To this end, we propose to finetune

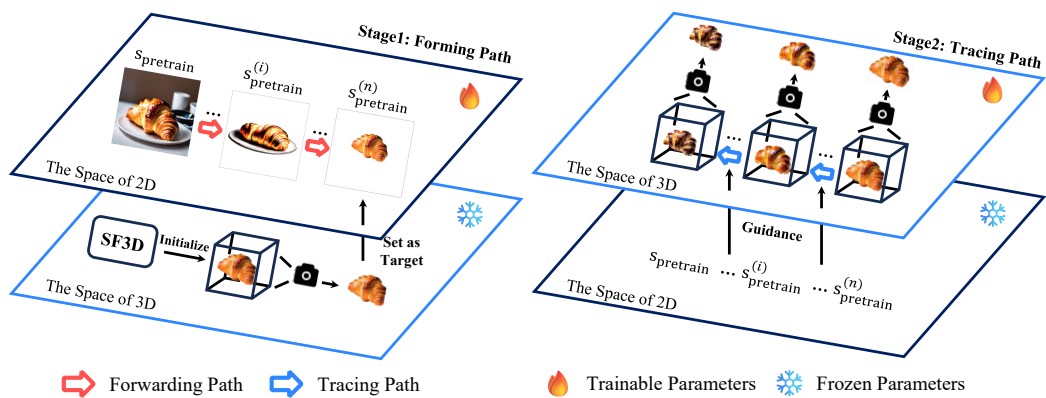

Figure 2: The illustration of our proposed path-tracing distillation. Given a text prompt, a 3D model is first initialized with SF3D (Boss et al., 2024) and gives several raw rendered images. In Stage1, these images are used to finetune the pretrained score network $s_{\text{pretrain}}$, and checkpoints $s_{\text{pretrain}}^{(i)}$ are saved to form a transformation path. In Stage 2, these checkpoints are used in a reversed order as the optimization targets giving score to optimize the 3D model.

$s_{\text{pretrain}}$ with rendered images to degrade the distribution and save intermediate checkpoints. Formally, given $s_{\text{pretrain}}$ parameterized by $\psi$ and an initialized 3D representation by $\theta$, $\boldsymbol{x}_0^\theta$ is a rendered image of a view $c$ and a prompt $y$. Then the finetuning (degrading) process is defined by

$$\min_{\psi} \mathbb{E}_{\boldsymbol{x}_0^\theta, y, t\sim\mathcal{U}(0,1), \epsilon\sim\mathcal{N}(0,1), c\sim p(c)} \left[ \|\epsilon_{\text{pretrain}}(\boldsymbol{x}_t^\theta, t, c, y; \psi) - \epsilon\|_2^2 \right]. \tag{6}$$

Here we adopt the noise prediction strategy (Ho et al., 2020) to maintain optimization stability. Throughout this learning phase, we preserve checkpoints of the intermediate transformations $\{s_{\text{pretrain}}^{(i)} | i = 0, 1, 2, \ldots, n\}$ within a predefined interval. Particularly, $s_{\text{pretrain}}^{(0)}$ is equal to $s_{\text{pretrain}}$ and $s_{\text{pretrain}}^{(n)}$ represents the distribution of $\boldsymbol{x}_0^\theta$ expectedly. Others are intermediate score networks specifically retained for path tracing processes in the following stage.

We make several strong assumptions in the forming path stage. The first assumption is the implicit connection between the parameter space of Stable Diffusion and the distribution space. This means though not converged, each checkpoint (a vector field) is assumed to represent a score network (a gradient field) of an unknown non-existent distribution. The second assumption is the steepest descent or traversal in the parameter space builds a reasonable path in the distribution space. The third assumption is the distance metric in the parameter space is translated to the similarity in the distribution space. Such assumptions require further exploration to justify, but this is beyond the scope of text-to-3D generation.

Several implementation details are introduced. We use Stable Fast 3D (SF3D) (Boss et al., 2024) to initiate the 3D Gaussian $\theta$, ensuring a plausible geometric and appearance configuration from text prompts. We use Stable Diffusion 2.1 as the backbone and the finetuing is performed with LoRA Hu et al. (2021) for parameter efficiency. Therefore, the saved checkpoints are only LoRA parameter attached on the backbone. The finetuning is done in only 400 iterations, which requires trivial extra computational cost.

### 4.3 TRACING PATH

This stage is to optimize the 3D model with the trasformation path. Existing models (Wang et al., 2023; Liang et al., 2024) uses only the $s_{\text{pretrain}}$ as the solitary target. In our design, the models $\{s_{\text{pretrain}}^{(i)} | i = 0, 1, 2, \ldots, n\}$ are employed in a reverse order replacing $s_{\text{pretrain}}$ to predict the scores of the noisy images $\boldsymbol{x}_t^\theta$. We iteratively update the 3D Gaussian $\theta$ by back-propagating the gradient of

the score discrepancy using ISM (Liang et al., 2024):

$$\min_{\theta \in \Theta} \mathbb{E}_{t \sim \mathcal{U}(0,1), c \sim p(c)} \left[ \omega(t) \| \epsilon_{\text{pretrain}}^{(i)}(\boldsymbol{x}_t, t, c, y) - \epsilon_{\text{pretrain}}^{(i)}(\boldsymbol{x}_s, s, c, \emptyset) \|_2^2 \right], \tag{7}$$

Here $\Theta$ is the space of $\theta$ with the Euclidean metric, $\omega(t)$ is a time-dependent weighting function and the noise prediction network $\epsilon_{\text{pretrain}}^{(i)}$ is used for approximating the score $s_{\text{pretrain}}^{(i)}$ by $s_{\text{pretrain}}^{(i)}(\boldsymbol{x}_t, t) \approx -\epsilon_{\text{pretrain}}^{(i)}(\boldsymbol{x}_t, t)/\sigma_t$.

During this process, to mitigate the Out-Of-Distribution (OOD) issue between $s_{\text{pretrain}}$ and generated samples, we use the penultimate score network $s_{\text{pretrain}}^{(n-1)}$ as target instead of $s_{\text{pretrain}}$ so that $s_{\text{pretrain}}^{(n-1)}$ can predict the score of generated samples more precisely. As the 3D Gaussian $\theta$ gets trained more convergent, we switch the target from $s_{\text{pretrain}}^{(n-1)}$ to $s_{\text{pretrain}}^{(n-2)}$, and then iteratively towards $s_{\text{pretrain}}^{(0)}$ $i.e.$ $s_{\text{pretrain}}$. The switch occurs every several optimization steps but ensure noisy images tend to lie in the distribution of $s_{\text{pretrain}}$ avoiding the OOD issue. Throughout the stage of tracing path, the 3D Gaussian $\theta$ is optimized with the guidance of the 2D score networks along the reversed transformation path.

Empirically, we find the transformation path Eq. 6 is not smooth, and changing LoRA target in this path frequently leads to the 3D shape and appearance collapse, as detailed in Appendix I. To get a more smooth path, we explore a more feasible strategy to implement path tracing. We only use the final LoRA checkpoint instead of the intermediate ones which save more disk memory and loading time. To avoid the shift of target problem, we provide the tracing path using **weighted LoRA**, which means the intermediate score networks are the interpolations between the zero LoRA (the pretrained score network) and the final LoRA with a coefficient $\boldsymbol{w} \in [0, 1)$. In the case, the targets in path tracing is obtained by forming an intermediate score network by varying $\boldsymbol{w}$ from 1 to 0. The optimization with LucidDreamer in this case is

$$\min_{\theta \in \Theta} \mathbb{E}_{t \sim \mathcal{U}(0,1), c \sim p(c)} \left[ \omega(t) \| \epsilon_{\text{pretrain}}^{\boldsymbol{w}}(\boldsymbol{x}_t, t, c, y) - \epsilon_{\text{pretrain}}^{\boldsymbol{w}}(\boldsymbol{x}_s, s, c, \emptyset) \|_2^2 \right], \tag{8}$$

Such variant promises a smooth enough and two-target transformation path. The final result in this case is less susceptible to the switch timing between intermediate score networks. The only parameter we need to control is the weight of the final LoRA, enhancing convenience and flexibility.

## 5 RELATED WORK

### 5.1 DIRECT 3D SHAPE GENERATION

Training neural networks using 3D models labeled with text description is a highly intuitive approach to text-to-3D model, which meets people's demand for more controllable 3D generation. For such pre-trained models, a 3D shape (Gao et al., 2022; Gupta et al., 2023; Nichol et al., 2022; Jun & Nichol, 2023; He et al., 2024) or 4D motion (Dabral et al., 2023; Zhang et al., 2023; Kim et al., 2023) can be inferred within minutes or seconds by inputting text prompt. Relevant research includes Point-E (Nichol et al., 2022), Shap-E (Jun & Nichol, 2023), Mofusion (Dabral et al., 2023), etc. Despite the high efficiency of these methods, the quality of outputs is often suboptimal, which is primarily due to the fact that the scale of 3D datasets cannot compare with that of 2D datasets. There are also works to improve quality of direct 3D shape generation by introducing novel frameworks and using 3DGS (Kerbl et al., 2023) as way of representing 3D shapes (He et al., 2024).

### 5.2 OPTIMIZING 3D SHAPE WITH 2D PRIORS

Given the typically small size of 3D model training sets, applying the rich 2D knowledge stored in pre-trained 2D models to the generation of 3D shapes has become a popular topic (Chen et al., 2024b; Jiang et al., 2023; Lorraine et al., 2023; Seo et al., 2023; Song et al., 2023; Yu et al., 2023a). Dream3D (Xu et al., 2023a) employs the pre-trained CLIP (Radford et al., 2021a) model and explicit three-dimensional shape priors to ensure that the rendered images have high semantic similarity to the given text prompts. DreamFusion (Poole et al., 2022) proposed Score Distillation Sampling (SDS), which optimizes 3D shapes using a 2D diffusion model. Subsequently, the introduction of CSD (Yu et al., 2023b) highlighted that the effectiveness of SDS stems from distilling knowledge from an implicit classifier rather than relying on generative priors. ProlificDreamer (Wang et al., 2023)

---

**Algorithm 1** Stage One: Forming Path

---

1: **Input:** 3D Gaussian $\theta$, SF3D initialized $\theta_{\text{SF3D}}$, score prediction network $s_{\text{pretrain}}$
2: **Output:** Path checkpoints $\{s_{\text{pretrain}}^{(i)}|i = 0, 1, 2, \ldots, n\}$
3: $\theta \leftarrow \theta_{\text{SF3D}}$
4: **while** Not Converged **do**
5:     $\boldsymbol{x}_0 = g(\theta, c)$
6:     $\boldsymbol{x}_t = \alpha_t \boldsymbol{x}_0 + \sigma_t \epsilon$
7:     Optimize **LoRA** with Eq.(6): $\min_\psi \mathbb{E}_{t,\epsilon,c} \left[ \|\epsilon_{\text{pretrain}}(\boldsymbol{x}_t^\theta, t, c, y; \psi) - \epsilon\|_2^2 \right]$.
8:     Save path checkpoints $s_{\text{pretrain}}^{(i)}$
9: **end while**

---

---

**Algorithm 2** Stage Two: Tracing Path

---

1: **Input:** 3D Gaussian $\theta$, score prediction network $s_{\text{pretrain}}$, $s_{\text{pretrain}}^{\text{LoRA}}$, LoRA load strategy constant $C$
2: **Output:** Well convergent $\theta$
3: **Define:** Function to calculate weight:
4:     $\text{weight}(iter) = \begin{cases} 1.0 - \left(\frac{iter-1}{1000}\right)^C, & \text{if } iter \leq 1000 \\ 0, & \text{otherwise} \end{cases}$
5: **while** Not Converged **do**
6:     $\boldsymbol{x}_0 = g(\theta, c)$
7:     Calculate current weight: $\boldsymbol{w} = \text{weight}(iter)$
8:     Optimize $\theta$ with Eq.(8):
9:         $\min_{\theta \in \Theta} \mathbb{E}_{t,c} \left[ \omega(t) \|\epsilon_{\text{pretrain}}^{\boldsymbol{w}}(\boldsymbol{x}_t, t, c, y) - \epsilon_{\text{pretrain}}^{\boldsymbol{w}}(\boldsymbol{x}_s, s, c, \emptyset)\|_2^2 \right]$.
10: **end while**

---

introduced Variational Score Distillation (VSD) to address the mode-seeking issues associated with SDS. PlacidDreamer (Huang et al., 2024) introduced Balanced Score Distillation (BSD) decomposing the SDS to avoid over-smoothing and over-saturation issues. Numerous other studies, such as DreamAvatar (Cao et al., 2023), improving the network design for human-related 3D generation, are dedicated to addressing the problems of low consistency and controllability of 3D model generation in specific generative jobs. Single-view construction and multi-view generation are also other ways of using 2D priors (Long et al., 2023; Hu et al., 2023; Lin et al., 2023; Liu et al., 2023b;a; Qian et al., 2024; Shi et al., 2023; Tang et al., 2023). Considering that there is currently no method capable of generating 3D models of sufficient quality for industrial applications, some research has shifted its focus to texture generation (Chen et al., 2024a).

## 6 EXPERIMENT

### 6.1 RESULT

We qualitatively compare the proposed method PTD with Magic123 (Qian et al., 2024), Fantasia3D (Chen et al., 2023), ProlificDreamer (Wang et al., 2023), and LucidDreamer (Liang et al., 2024) (Figure 3). Images of other works are sourced from LucidDreamer (Liang et al., 2024). Results demonstrate the superior appearance and texture fidelity achieved by our method. In terms of generation speed, our approach employs the same architecture as LucidDreamer, ensuring no additional time is required. More generated results are in Figure 1.

We also quantitatively evaluate our model (Table 1) with CLIP similarity. We compute CLIP similarity based on 415 prompts from DreamFusion (Poole et al., 2022). CLIP similarity is calculated using OpenAI's ViT-L/14 (Radford et al., 2021b) and OpenCLIP's ViT-bigG-14 (Schuhmann et al., 2022). During computation, we set a camera radius of 4, elevations of 0 and 45 degrees, and select 8 evenly

*Magic3D* (~6h)    *Fantasia3D* (~6h)    *ProlificDreamer* (~8hs)    *LucidDreamer* (~1.5h)    ***Ours*** **(~1.5h)**

*"A DSLR photo of the Imperial State Crown of England."*

*"A DSLR photo of a Schnauzer wearing a pirate hat."*

Figure 3: Comparison of 3D generation quality across different methods. Our method enjoys similar time cost to LucidDreamer but has better visual quality.

spaced azimuth angles every 45 degrees, ranging from -180 to 180 degrees. For each prompt, we render 16 images to compute CLIP similarity. The comparison is done with DreamFusion (Poole et al., 2022), Instant3D (Li et al., 2023), ProlificDreamer (Wang et al., 2023), and LucidDreamer (Liang et al., 2024). Results of these methods are sourced from GaussianDreamer (Yi et al., 2024). In this comparison, our method demonstrates higher congruence between text and images, indicating superior image quality.

## 6.2 GENERALIZABILITY OF PATHTRACING DISTILLATION

To evaluate the generalizability of PathTracing Distillation method, we compare the ISM (Liang et al., 2024) and our method in different 3D representations. In Figure 4, we generate the 3D asets with the same prompts in 3DGS and NeRF representations respectively. As for the outcomes of other work, we borrow figures from LucidDreamer. We can intuitively see that the 3D assets our method generate is more detailed and have more diversity. Even in NeRF representation, our methods also work well to generate 3D assets with vivid appearance and good shape.

## 6.3 ABLATION STUDY

**Effect of PathTracing Distillation** We compare our Path Tracing Distillation (PTD) method with the vanilla method in Figure 5. Additionally, we evaluate different rates of LoRA weight change and perform an ablation study on the Path Tracing component. The results demonstrate that our PTD method achieves better convergence within the same number of training steps.

Table 1: Quantitative comparison of the proposed PTD with other methods. The best performance is highlighted in bold and the second best is underlined. Our method outperforms other methods with an average generation time cost of 1.5 hours. Our results are comparable with ProlificDreamer which takes about 8 hours, over 5 times of temporal cost as ours.

| Methods | ViT-L/14 ↑ | ViT-bigG-14 ↑ |
|---|---|---|
| DreamFusion | 23.60 | 37.46 |
| ProlificDreamer | **27.39** | **42.98** |
| Instant3D | 26.87 | 41.77 |
| LucidDreamer | 25.99 | 40.27 |
| Ours | 27.16 | 42.36 |

**Initialization with SF3D Generator** 3D Gaussian is sensitive to the initialization of shape and appearance. We ablate SF3D method (Boss et al., 2024) and use PointE (Nichol et al., 2022) instead to validate the advantage of using SF3D. In Figure 6, we can see a higher text and image matching when using SF3D, indicating a better initial shape and appearance. This illustrates that SF3D method could help to produce a better outcome.

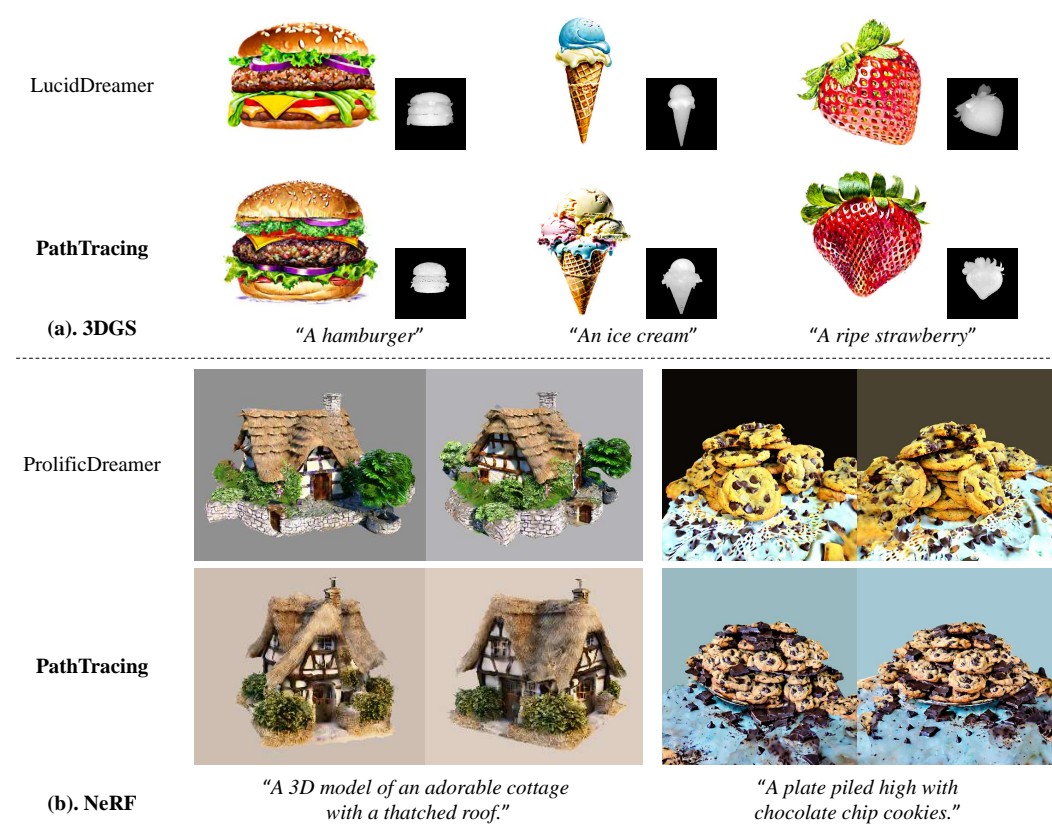

Figure 4: Generalization of PathTracing Distillation. When accompanied with 3DGS or Nerf, our proposed PathTracing Distillation can both achieve appealing visual quality.

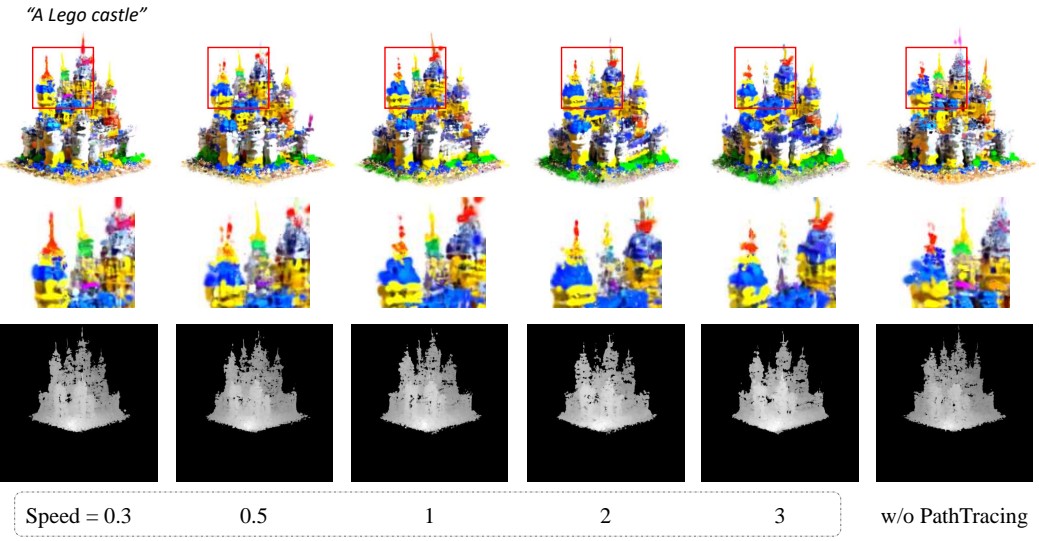

Figure 5: Ablation study of the Path Tracing Distillation (PTD) method. The figure compares the convergence results of our PTD method with the vanilla method at the same training step. Different rates of LoRA weight change are also evaluated. It is evident that our PTD method achieves better convergence in detail.

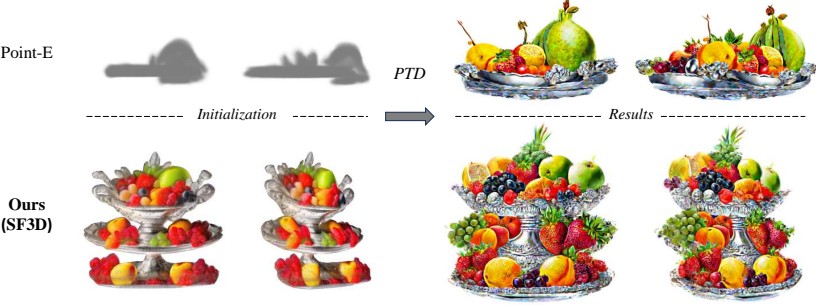

Figure 6: The ablation of SF3D. The initialization with SF3D is beneficial to final generated results.

**Effect of Finetuned Pretrained Model**  We need to use the pretrained model to generate the the whole described object. However the Stable Diffusion 2.1 model (Rombach et al., 2022a) tend to generate images with cropped objects due to the dataset it used to train is through data augmentation. Thus, we finetune the base model as the pretrained model with high quality images from Stable Diffusion XL (Podell et al., 2023), ensuring the generation of complete objects. We ablate the finetuned step and evaluate the outcome. From the Figure 7, we can compare the visual results. With the finetuned model, the final 3D Gaussian shows more detailed appearance and complete shape.

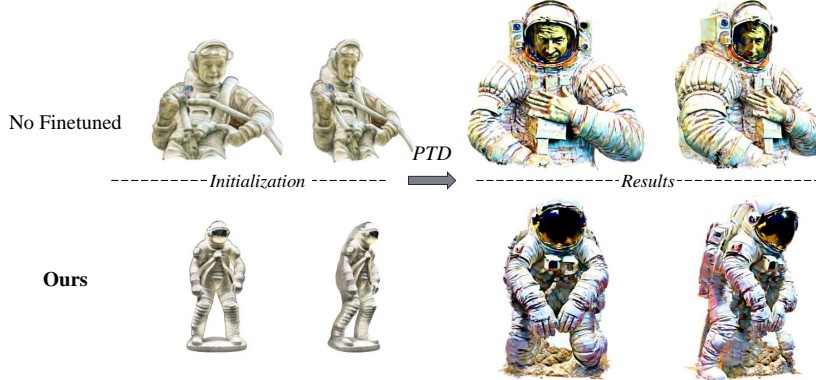

Figure 7: Ablation of Finetune the pretrained score network. To avoid giving guidance of cropped objects from Stable Diffusion 2.1, we finetune it with images of whole objects generated by Stable Diffusion XL. Such finetuning helps in getting complete 3D models.

## 7  CONCLUSION

In this paper, we discuss the OOD issue in the current text-to-3D method with score distillation. We find rendered images from the initial 3D models lie out of the distribution of the pretrained score network, which is typically Stable Diffusion. Such issue cause unstable score prediction for the generation process. To solve this issue, we propose a path tracing method. Experimental results demonstrate that our proposed method enhances the generation process, achieving higher CLIP similarity and maintaining visual quality.

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

## A  LIMITATION AND POTENTIAL SOCIAL IMPACT

Our paper relies on the assumption that the intermediate checkpoints during the forwarding stage are score functions representing existing or non-existent distributions. However, these checkpoints at least only represent vector fields, and a vector field is a gradient field when several conditions are satisfied, including path independence, continuous partial derivatives, and zero curls (Matthews, 1998). The violation may hinder the optimization in the generation process.

The claim of OOD issue may not be applicable when the pretrained score network $s_{\text{pretrain}}$ includes low-quality images similar to initial rendered images. Our claim is valid as current methods adopt Stable Diffusion as $s_{\text{pretrain}}$, which is trained on high-quality realistic images to form a 2D prior.

The proposed method introduces extra computational cost as in the forward path stage. However, empirically we find the cost is limited, but the path significantly reduce the cost in the path tracing stage. The generated 3D models reach nearly the same quality within 400 steps as those generated by Gaussian Dreamer with 1000 steps (Figure 1). The extra computational cost is balanced.

This paper aims to improve the generation speed of generative models, which could be used to generate fake 3D model for disinformation.

## B    ETHICS STATEMENT

In this study, human subjects participated in a user study aimed at evaluating the quality of 3D models generated by our method compared to an existing method. We recruited five volunteers who were thoroughly informed about the study's purpose, procedures, and any potential risks prior to participation, ensuring that they provided their informed consent freely. Participants were briefed on the ability to withdraw from the study at any time without negative consequences, underscoring our commitment to their autonomy and well-being. During the study, each volunteer assessed 8-second videos of 3D models generated from 300 different prompts. Following their involvement, we conducted a debriefing session to address any discomfort or questions, ensuring that participants felt supported and valued. Privacy and confidentiality were maintained rigorously; no personally identifiable information was collected, and responses were anonymized. All data were securely stored in compliance with data protection regulations and were used exclusively for the research. We identified no additional ethical concerns, such as bias, privacy, or conflicts of interest, in the course of our study, affirming our adherence to ethical research standards and practices.

## C    REPRODUCIBILITY STATEMENT

In our study on the 3D generation task, we identified an Out of Distribution (OOD) issue, validated in Section 3. We provide an additional validation experiment in Appendix H to assess the mitigation of this issue. To address the OOD problem, we propose a PathTracing Distillation method. Detailed implementation information is available in Appendix E. The code is included in the supplemental material and will be released shortly.

## D    RELATIONSHIP WITH PREVIOUS METHODS

**The connection to GAN**    Score distillation is connected to Generative Adversarial Networks (GANs). The adversarial discriminator is also used in 3D reconstrucition (Roessle et al., 2023). In this case, GAN does not suffer from the OOD issue. The reason lies in the difference between the adversarial training and diffusion training. Given the real image distribution $q$ and the rendered image distribution $p$, the KL divergence follows

$$D_{\mathrm{KL}}(q \parallel p) = E_q \left[ \log \frac{q}{p} \right] \tag{9}$$

An adversarial discriminator $D(\cdot)$ approximates $D(x) = \frac{q(x)}{q(x)+p(x)}$, and the connection between the adversarial discriminator and the scores can be summarized as Luo et al. (2023)

$$\log \frac{D(x)}{1 - D(x)} = \log \frac{q(x)}{p(x)} = \log q(x) - \log p(x) \approx s_{pt}(x) - s_\phi(x) \tag{10}$$

Therefore, both GAN and score distillation minizes the KL divergence to form the final 3D model. However, in the adversarial training, the discriminator $D$ is directly trained on both real and generated data jointly, and it enjoys a support of both distribution but may suffer from gradient saturation (Arjovsky et al., 2017). On the other hand, in score distillation the networks $s_{pt}(x)$ and $s_\phi(x)$ are trained separately and they have different support. Therefore the score prediction is unstable in this case.

**Curriculumn Learning**    Our path tracing method shares a similar spirit with Curriculumn Learning. Curriculum learning (CL) (Wang et al., 2022; Bengio et al., 2009) is a training strategy designed to

improve the learning process of machine learning models by organizing data in a way that mimics the natural progression of complexity and difficulty, similar to how humans learn. This approach involves training a model on easier data before moving to harder data, thereby enhancing the model's generalization capacity and convergence rate.

The transformation path constructed in our method is indeed a curriculum with easier distributions at the beginning of generation but harder distributions at the end. Specifically, those checkpoints in the transformation path close to $s_{pt(n)}$ are easy distributions as they are nearly converged to the render image distribution. Those checkpoints close to $s_{pt}$ are difficult distributions as they are quite different from the rendered image distribution and it may take quite a long time to learn. The target to learn is iterated during the training similar to the adaptive curriculum learning (Kong et al., 2021).

## E  IMPLEMENTATION DETAILS

Our approach is meticulously crafted in PyTorch, leveraging the foundational structures of Lucid-Dreamer (Liang et al., 2024). We fintune the "stabilityai/stablediffusion-2-1-base" with the LoRA rank of 4. The Unet is trained only with a learning rate of 0.0001. The LoRA used for forming path is the same configuration. We train the finetune-LoRA with 400 epochs and formaing-path-LoRA with 200epochs. The LoRA load strategy we choose is 0.3. We use the PathTracing Distillation in the first 1000 steps.

For the 3D Gaussian initialization, we initiate from a SF3D (Boss et al., 2024)-initialized point cloud. Other configuration is the same with LucidDreamer. All experiments are executed on RTX 3090.

## F  TWO TYPES OF OUT OF DISTRIBUTION ISSUES

In the context of 3D generation, there are two primary types of out-of-distribution (OOD) issues. We have discussed one of it in the main context. Another type is associated with the inaccuracy of the score prediction network $s_\phi$ in predicting the score from the noisy rendered images $x_t$ to the clear ones $x_0$. This type of OOD problem often arises in works that employ two networks to optimize the 3D representation, such as ProlificDreamer (Wang et al., 2023), where one network $s_\phi$ is used to predict the score of noisy rendered images. It also has a close relationship with the initialization strategy of the score prediction network $s_\phi$ which typically follows by three approaches:

1. **Random Initialization**: This method initializes $s_\phi$ without any prior information, which often results in $s_\phi$ being initially distant from the distribution of rendered images, leading to inaccurate score predictions in the early stages.

2. **LoRA Initialization**: In this approach, $s_\phi$ is initially set as a Low-Rank Adaptation (LoRA) of a pretrained model $s_{\text{pretrain}}$, endowing it with robust prior knowledge. At the beginning of the training, the elements in the LoRA matrix $B$ are zero, making $s_\phi$ identical to the pre-trained model. As training progresses, $s_\phi$ gradually adapts to fit the distribution of the rendered images. Notably, ProlificDreamer employs this initialization strategy. However, due to the initial suboptimal quality of the rendered images, $s_\phi$ fails to predict the score accurately, presenting another OOD challenge.

3. **Pre-training with 3D Model**: Before training the 3D model, $s_\phi$ is initialized by fitting it to the images rendered from the initial 3D model. This method ensures that $s_\phi$ starts with a distribution that closely matches the rendered images, facilitating accurate score predictions from the onset. Similar to the second method, during the subsequent training process, the optimization of the 3D model and $s_\phi$ alternates to ensure $s_\phi$ can consistently predict the noise accurately.

In methods using two score networks, the third initialization approach is necessary and advantageous as it aligns $s_\phi$'s distribution with that of the 3D model-rendered images, thereby ensuring accurate and stable score predictions throughout the 3D generation process.

## G  SCORE APPROXIMATED THROUGH THE STABLE TARGET FIELD METHOD

To empirically investigate the OOD issue, we compare the predicted score $s_{\text{pretrain}}(\boldsymbol{x}_t, t, y)$ with the approximated score $s_{\text{approx}}(\boldsymbol{x}_t, \boldsymbol{x}_0^{\text{ref}})$ of rendered images $\boldsymbol{x}_t^{\theta}$ or generated images $\boldsymbol{x}_t^{\text{pretrain}}$. $\boldsymbol{x}_0^{\text{ref}}$ is the approximation required reference samples from the pretrained model. The rationale behind is that the score prediction network is trained to fit real scores by score matching (Vincent, 2011), and the matching loss is low when given in-distribution samples but high otherwise. The scores $s_{\text{real}}(\boldsymbol{x}_t)$, expectedly equaled to $\nabla_{\boldsymbol{x}_t} \log p_t(\boldsymbol{x}_t)$, is unknown but can be approximated through using Stable Target Field (STF) (Xu et al., 2023b). The approximation given by STF is to utilize the training data $\boldsymbol{x}_0^{\text{ref}}$ to perform a weighted sum of conditional score with reduced variance.

$$s_{\text{approx}}(\boldsymbol{x}_t, \boldsymbol{x}_0^{\text{ref}}) \approx \sum_{i=1}^{n} \frac{p_{t|0}\left(\boldsymbol{x}_t \mid \boldsymbol{x}_0^{\text{ref}(i)}\right)}{\sum_{j=1}^{n} p_{t|0}\left(\boldsymbol{x}_t \mid \boldsymbol{x}_0^{\text{ref}(j)}\right)} \nabla_{\boldsymbol{x}_t} \log p_{t|0}\left(\boldsymbol{x}_t \mid \boldsymbol{x}_0^{\text{ref}(i)}\right), \tag{11}$$

where $n$ is the number of reference samples. Notice that the approximation does not use any network prediction.

## H  OUT OF DISTRIBUTION IN TRAINING 3D

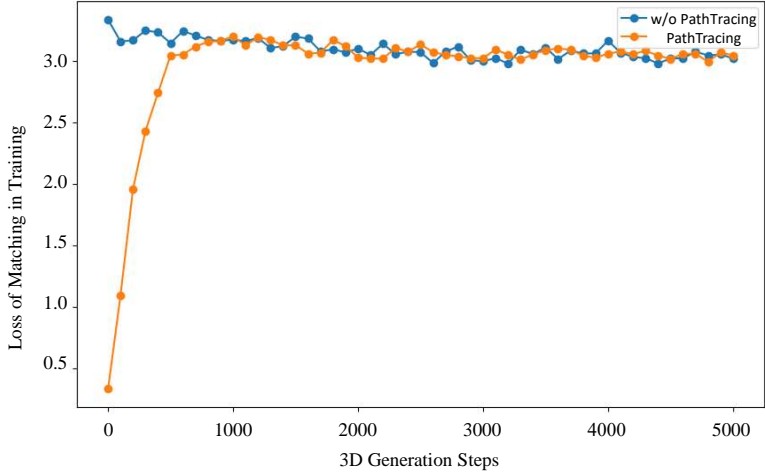

Figure 8: The illustration of the matching loss given vanilla method and PathTracing Distillation (PTD) method in training. The loss of the vanilla method in blue is significantly higher than the loss of PTD method in green until closing to the end of training, indicating the score inaccurately predicted due to rendered images lying outside of the pretrained score network distribution in the vanilla method. Therefore, the PTD method could effectively help the stable 3D training.

In this section, we validate the effect of OOD mitigation using PathTracing Distillation (PTD). In Figure 8, we can intuitively observe the mitigation of OOD issue by plotting the curve of $\mathcal{L}_{\text{matching}}$ in the PTD method over the course of training 3D. $\boldsymbol{x}_0^{\theta}$ are images rendered from 3D Gaussian $\theta$ in different steps of training and $\boldsymbol{x}_0^{\text{ref}}$ are images sampled from the guidance model. We use the specific guidance corresponding to training steps to calculate the $\mathcal{L}_{\text{matching}}$, where it is the $s_{\text{pretrain}}^{\boldsymbol{w}\text{LoRA}}$ in our PTD method and $s_{\text{pretrain}}$ in the vanilla method. Since the $\mathcal{L}_{\text{matching}}$ of the vanilla method decline along with training steps and the $\mathcal{L}_{\text{matching}}$ of the PTD method is always lower, we can conclude that the following inequality holds:

$$\mathbb{E}_t \| s_{\text{pretrain}}^{\boldsymbol{w}\text{LoRA}}(\boldsymbol{x}_t, t, y) - s_{\text{approx}}(\boldsymbol{x}_t) \|_2 \leq \mathbb{E}_t \| s_{\text{pretrain}}(\boldsymbol{x}_t, t, y) - s_{\text{approx}}(\boldsymbol{x}_t) \|_2. \tag{12}$$

Therefore, we can validate that the PTD method effectively reduce the OOD issue.

## I    SHIFT OF LORA IN TRAINING 3D

The traditional approach of changing LoRA target is to use the intermediate checkpoint of LoRA in training. But three drawbacks determine its unconveniency:

1. **Shift of Targets**: Over the course of training 3D, a stable and solid target pretrained model is important. If we change the LoRA of pretrained model just like changing a accessory, the final target $\theta$ is also changing even though the Exponential Moving Average (EMA) trick is used to train smooth LoRA. Reflected to current $\theta$, the appearance and shape we construct in current LoRA is destroyed when changing to the next one (Huang et al., 2024).

2. **Limited Number of Checkpoints and Difficulty of Selecting Strategy of Loading LoRA**: We always train LoRA using different batch size, dataset size and some other parameters. In some extreme situations, we only train several epochs which is so short to produce to provide feasible path. In addition, with some intermediate LoRAs, we may be confused to select the suitable interval to change LoRA and the LoRA step gap. It always needs parameter search which will consume a lot of time.

3. **Resources consumed**: The intermediate LoRA checkpoints are needed to be saved locally, requiring extra disk memory. And loading new LoRA and setting configuration consume time. Both need resources to implement.

Base on these reasons, we adopt the new weighted LoRA method to trace path, showed in Section 4.3.

## J    MORE RESULTS

In Figure 9, we present additional generation results obtained using our PathTracing Distillation methods. These examples illustrate the versatility and efficacy of our approach in handling various object categories. The generated results showcase high-quality renderings across multiple domains, such as animals, plants, machinery, and food items. By closely examining the details, the robustness and precision of our framework in capturing the intricate features and appearances of different objects can be appreciated. The success in these diverse categories underscores the broad applicability and effectiveness of our method.

## K    FAILURE EXAMPLE

During the generation process, occasional failures can occur. As illustrated in the Figure 10, we present three examples stemming from different causes. The left image represents a case where the attributes of a 3D object were not correctly matched with the corresponding descriptive terms; for instance, the attribute "silver" was incorrectly applied to cheese instead of the intended plate, resulting in an erroneous sample. The middle image demonstrates the difficulty in accurately generating all objects and their corresponding attributes in the context of complex prompts. The right image depicts a scenario where the quantity of objects was incorrectly generated, leading to the omission of one object.

## L    MORE APPLICATIONS

**Image to 3D Generation**    In this section, we expand our pipeline to some new application, showed in Figure 11(a). Since we use SF3D (Boss et al., 2024) for initialization of 3D Gaussian, we could also use it to reconstruct 3D Gaussian from images. Given a single image used for reconstruction, we automatically remove the background and get the image with only central objects. Then, a mesh will be reconstructed and we could use it to initialize the 3D Gaussian. With respective prompts, we transform it into vivid 3D Gaussian.

**Mesh to 3D Generation**    Mesh is a popular representation for 3D. We could also use our pipeline to add more details or change style base on given coarse mesh. With the control prompt, the PathTracing Distillation method change it into different appearance and shape, as showed in Figure 11(b).

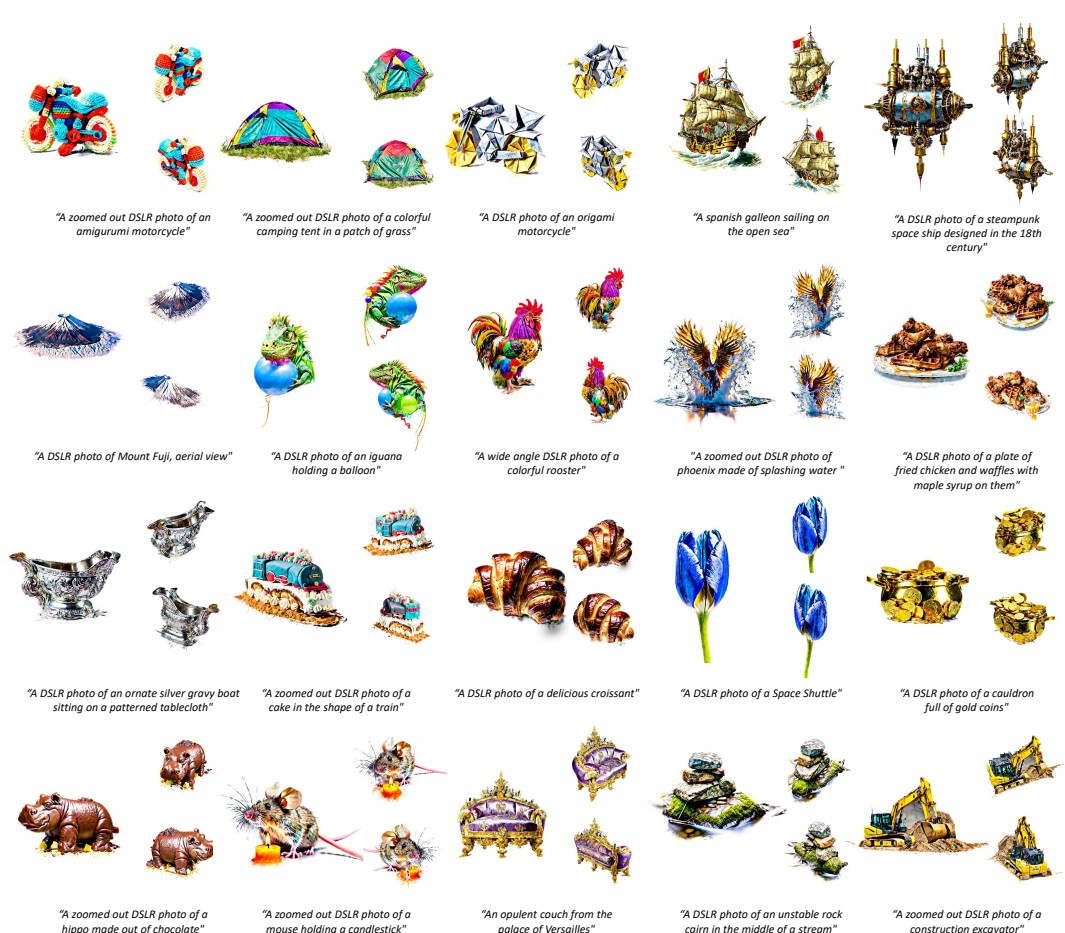

*"A zoomed out DSLR photo of an amigurumi motorcycle"*   *"A zoomed out DSLR photo of a colorful camping tent in a patch of grass"*   *"A DSLR photo of an origami motorcycle"*   *"A spanish galleon sailing on the open sea"*   *"A DSLR photo of a steampunk space ship designed in the 18th century"*

*"A DSLR photo of Mount Fuji, aerial view"*   *"A DSLR photo of an iguana holding a balloon"*   *"A wide angle DSLR photo of a colorful rooster"*   *"A zoomed out DSLR photo of phoenix made of splashing water "*   *"A DSLR photo of a plate of fried chicken and waffles with maple syrup on them"*

*"A DSLR photo of an ornate silver gravy boat sitting on a patterned tablecloth"*   *"A zoomed out DSLR photo of a cake in the shape of a train"*   *"A DSLR photo of a delicious croissant"*   *"A DSLR photo of a Space Shuttle"*   *"A DSLR photo of a cauldron full of gold coins"*

*"A zoomed out DSLR photo of a hippo made out of chocolate"*   *"A zoomed out DSLR photo of a mouse holding a candlestick"*   *"An opulent couch from the palace of Versailles"*   *"A DSLR photo of an unstable rock cairn in the middle of a stream"*   *"A zoomed out DSLR photo of a construction excavator"*

Figure 9: Additional examples generated by our PathTracing framework. The results demonstrate the framework's capability to produce high-quality renderings across a diverse range of object categories, including animals, plants, machinery, and food items. Please zoom in for a detailed view.

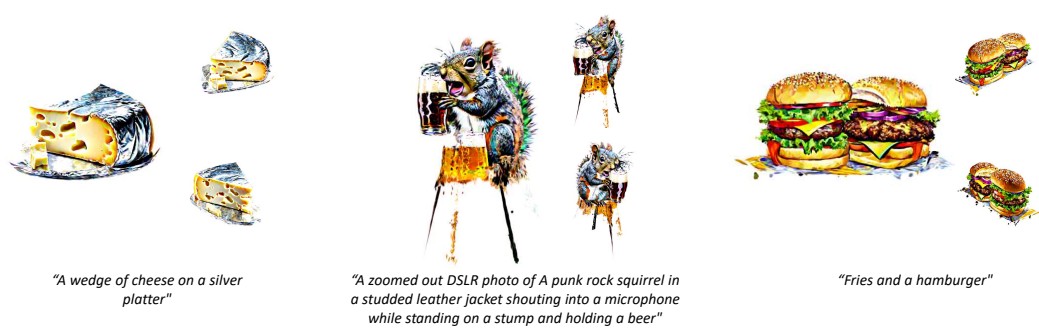

*"A wedge of cheese on a silver platter"*   *"A zoomed out DSLR photo of A punk rock squirrel in a studded leather jacket shouting into a microphone while standing on a stump and holding a beer"*   *"Fries and a hamburger"*

Figure 10: Examples of failures during the generation process. The left image shows incorrect attribute matching, where the silver attribute is assigned to cheese instead of a plate. The middle image illustrates the challenge of accurately generating all objects and their descriptions in complex prompts. The right image depicts an error in the number of generated objects, with one object missing.

**(a). Image to 3D:**

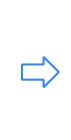 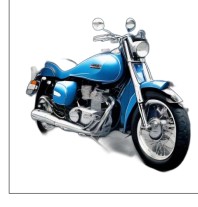 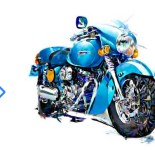 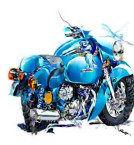 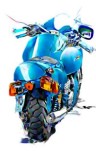

*"A brightly colored mushroom growing on a log"*

**(b). Coarse Mesh to Detailed 3D:**

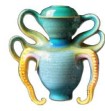 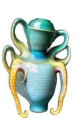 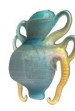 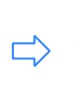 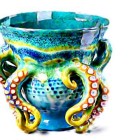 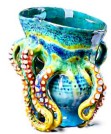 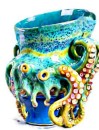

*"A ceramic upside down yellow octopus holding a blue green ceramic cup"*

Figure 11: Demonstrations of our pipeline's applications: (a) Image to 3D Generation showcases the process where a single image is used to reconstruct a 3D Gaussian representation by first isolating the central object and then transforming it into a detailed 3D model. (b) Coarse Mesh to Detailed 3D Generation illustrates how our method can enhance or restyle a coarse mesh into a more detailed and visually different 3D model through PathTracing Distillation. These examples highlight the versatility and effectiveness of our pipeline in various 3D generation tasks.

