# OpenReview forum: "Path-Tracing Distillation: Enhancing Stability in Text-to-3D Generation by Mitigating Out-of-Distribution Issues"
_ICLR.cc/2025/Conference — ICLR 2025 Conference Withdrawn Submission_

### Official Review · Reviewer_A18m · 2024-10-27

**Soundness:** 3
**Presentation:** 3
**Contribution:** 2
**Rating:** 3
**Confidence:** 5

**Summary:**

This paper proposes path-tracing distillation, a method for text-to-3D generation. This work builds upon the Score Distillation line of work that uses pre-trained text-to-image models to distill knowledge into a 3D model using per-prompt optimization. First, a pre-trained feed-forward 3D generator is used to synthesize the initial 3D model with rendered images. Subsequently, the pre-trained score network is fine-tuned and checkpoints are saved during optimization. In the following stage the saved checkpoints are used in reverse order to optimize the 3D model. The work can synthesize diverse text-to-3D results.

**Strengths:**

- Writing quality: The paper is well written and the motivation is clear.
- Code available: This can help the community to build on this method.

**Weaknesses:**

-  Oversaturated results: The results demonstrate significantly less appearance realism than previous works such as ProlificDreamer. The quality looks similarly cartoonish as LucidDreamer and it is not clear where this is coming from. Most results look like paintings, e.g., in Fig. 1 the mouse playing a touba or the duck swimming look very oversaturated.
- Sparse evaluation: There are only some CLIP scores provided which mainly measure text alignment from a single 2D rendered image. There are no 3D metrics used to evaluate the approach. The bare minimum is to conduct a user study as in previous works.
- Two-stage process: Relying on a two-stage process where SF3D is used as an initialization kind of breaks the story of the paper. How does it work without that? Since the model will be mainly determined by feed-forward generation methods and now rather looks like a texture boosting method. Previous SDS methods generate assets from scratch without external models. The geometry of the final scene is basically determined by the feed-forward model SF3D, and these feed-forward models are mainly trained on synthetic data. Hence, they do not have good prompt alignment and are limited in their output which will project into this SDS line of work. The main advantage of SDS works is that they mainly use 2D priors and have much more potential for text alignment, scene complexity, and appearance quality. But this is somehow broken through using SF3D as an initialization.
- Requirement of multiple checkpoints: In the first stage of optimization many checkpoints are saved to fit the rendered images, this provides a large overhead. I do not fully understand why this can not be done on the fly without saving a lot of model checkpoints to disk, as it also breaks continuity and introduces discretization artifacts by setting a hyperparameter of how many checkpoints are feasible to store.
- No video results: For 3D tasks it is very important to include renderings in the form of videos for readers to judge the results. It is difficult to evaluate the method based on the paper alone.

**Questions:**

I currently rate this paper below the acceptance threshold. The main reasons are the mediocre results and missing evaluations, which are not very convincing. The results look similar to LucidDreamer, which is already worse than ProlificDreamer in terms of appearance quality. Currently, this work is rather making a step back towards oversaturation and the decrease in training time is not significant to justify it. I do not see the benefit of the method to the community in its current form.

I would like authors to address following questions:
- Why do the results look so saturated compared to ProlificDreamer?
- Why are there no comparisons which actually evaluate 3D?

I am open to adjusting my rating based on the rebuttal.

---

### Official Review · Reviewer_nvCm · 2024-10-29

**Soundness:** 2
**Presentation:** 2
**Contribution:** 2
**Rating:** 3
**Confidence:** 4

**Summary:**

This paper addresses the instability in SDS-based text-to-3D generation. The authors suggest that this instability arises from OOD: the rendered images from initial, low-quality 3D models don't fit within the training distribution of the pretrained 2D score network. To tackle this, they introduce Path-Tracing Distillation (PTD), a two-stage method designed to resolve the OOD problem by creating a "path" of intermediate score networks. In the first stage, a feed-forward 3D generation model initially generates the 3D content, and then rendered views are used to fine-tune the pretrained score model (diffusion). In the second stage, this updated score model is used in reverse as distillation targets, guiding the optimization of the 3D model. Experiments demonstrate that PTD enhances both the stability and quality of text-to-3D generation.

**Strengths:**

1: The paper identifies and empirically demonstrates the OOD problem as a source of instability in text-to-3D generation.

2: It shows the applicability of PTO across various 3D representations, such as 3DGS and NeRF, indicating a broader potential for the proposed method.

3: Overall the paper is easy to follow, idea is straightforward.

**Weaknesses:**

1: The paper's literature review could be more comprehensive, especially regarding recent studies on distribution-related  or initialization -related issues with SDS. A more thorough discussion of related work and clearer differentiation from existing approaches would enhance the paper's contribution. Relevant works include:

DreamMapping: High-Fidelity Text-to-3D Generation via Variational Distribution Mapping

Rethinking Score Distillation as a Bridge Between Image Distributions

Connecting Consistency Distillation to Score Distillation for Text-to-3D Generation

2: While the reported CLIP scores show improvements over some baselines, they are only comparable to those achieved by ProlificDreamer. This raises concerns about the practical significance of the proposed method. The paper lacks additional evidence, such as a user study or an anonymous demo project page, to demonstrate a clear advantage in terms of visual quality or perceptual realism.

3: The rationale for using SF3D for 3D Gaussian initialization is not fully convincing. SF3D utilizes DMTet as its 3D representation, and extracting point cloud from it to initialize 3DGS seems indirect and potentially inefficient. Alternative initialization strategies, such as directly using feed-forward Gaussian models, could be explored and compared. The paper lacks a thorough justification for the chosen approach.

**Questions:**

The ethics statement mentions human user studies, but no results are provided in the main paper or supplementary material. The ethics statement seems irrelevant to the paper's content?

Overall this paper introduces a reasonable approach with potential, but it requires a more comprehensive literature review, stronger empirical validation, and clearer justification of methodological choices to strengthen its contribution. My current recommendation is reject.

---

### Official Review · Reviewer_3gZm · 2024-10-30

**Soundness:** 2
**Presentation:** 3
**Contribution:** 2
**Rating:** 3
**Confidence:** 3

**Summary:**

The authors make efforts to identify and tackle the OOD issue in the current text-to-3D generation methods, especially during the early stages of 3D model generation. The authors propose two-stage approach, path-tracing distillation, to optimize the 3D model by using intermediate score networks. Experimental results showcase their approach enhances the 3D object generation with better visual quality.

**Strengths:**

1) Identify the OOD issue in the current text-to-3D generation methods with score distillation.
2) Propose the PTD technique to tackle the OOD issue, especially during the early stages of training.
3) Overall presentation is clear and easy to follow.

**Weaknesses:**

1) There is no theoretical support on the OOD issue as found out by the authors, though it is likely that the rendered images from initial 3D models are out of distribution of the high-quality images used to train the score model.
2) It is not clear whether the OOD issue during early stage of the training will affect the final results or not, at the late stage of the training, the 3D model will render better quality images and therefore align with the high-quality images, the OOD is not an issue anymore, the final result should not affect too much by the OOD in the early stage.
3) The experimental results is ok but not convincing, for example, Table.1 Prolific Dreamer outperforms yours though the authors claim the former is 5 times over yours, since the authors claim their approach can be integrated to most of existing methods, the best way to compare is for each methods in Table.1, you may come out two results, one for original method, the other for original method integrated yours. for another example, ablation study does not add any value, to me, it seems the authors just selected the best combination to achieve the best result, the approach may not general to various cases.
4) Minors: Figure 5 appears somewhat blurred; consider using a vector image. There are also several typos, such as "the the" throughout the paper. The terms "CLIP similarity" and "CLIP score" suddenly appear in the abstract, etc.

**Questions:**

It remains unclear whether the OOD issue during the early stages of training will affect the final results. As training progresses, the 3D model is expected to render higher quality images that align more closely with the high-quality training images, potentially mitigating the OOD concern. In such a way, the final results should be less / not impacted by OOD in the early stages.

---

### Official Review · Reviewer_KQsV · 2024-10-31

**Soundness:** 3
**Presentation:** 2
**Contribution:** 2
**Rating:** 3
**Confidence:** 4

**Summary:**

The authors propose a Score Distillation Sampling (SDS)-based text-to-3D generative model called Path-Tracing Distillation (PTD). They observe that raw rendered images from the initial 3D representation lie outside the distribution of the pretrained score prediction network, leading to instability. To address this, they refine the pretrained model by employing a series of score prediction networks instead of a single model. In the next stage, they use these score prediction networks to train their 3D representation.

**Strengths:**

- The motivation is well-established
- The approach of using a series of score prediction models instead of a single one, with negligible cost increase, is novel and interesting.

**Weaknesses:**

- The experimental details for understanding the out-of-distribution issue are missing. For example, the number of samples used to calculate $ s_{\text{approx}} $ and the specific text prompts are not provided. Additionally, while the authors mention that the difference between $ s_{\text{pretrain}} $ and $ s_{\text{approx}} $ is more pronounced when $ t $ is small, including plots showing this difference would better establish the motivation for the method.

- The authors claim their method outperforms other state-of-the-art methods; however, models trained with PTD are initialized by SF3D [1], while other methods are either randomly initialized or use Point-E [2]. SF3D is a stronger initialization, making it unclear whether the performance gain is due to the initialization or the path-tracing approach itself.

- To understand the specific contribution of path-tracing, the authors should compare PTD against LucidDreamer initialized by SF3D, which would help isolate the impact of PTD from the initialization effects.

- There are very few qualitative comparisons with other state-of-the-art methods. More qualitative examples would provide stronger visual evidence of PTD’s effectiveness.

References

[1] Mark Boss, Zixuan Huang, Aaryaman Vasishta, and Varun Jampani. Sf3d: Stable fast 3d mesh reconstruction with uv-unwrapping and illumination disentanglement, 2024. URL https:// arxiv.org/abs/2408.00653.

[2] Alex Nichol, Heewoo Jun, Prafulla Dhariwal, Pamela Mishkin, and Mark Chen. Point-e: A system for generating 3d point clouds from complex prompts. arXiv preprint arXiv:2212.08751, 2022.

**Questions:**

Please refer to the weakness. My main concern are about the influence of initialization and insufficient qualitative comparisons against SOTA methods.

---

### Official Review · Reviewer_XGRQ · 2024-11-04

**Soundness:** 2
**Presentation:** 2
**Contribution:** 2
**Rating:** 5
**Confidence:** 5

**Summary:**

This paper focuses on the enhancement of score distillation in text-to-3D generation. It begins by identifying a key limitation: images rendered from a 3D scene may diverge from the distribution learned by the pre-trained diffusion model, resulting in imprecise supervision for 3D representations. To overcome this, the paper introduces Path-Tracing Distillation, an approach that first fine-tunes the diffusion model on images rendered from a 3D scene generated by a zero-shot text-to-3D method. This fine-tuned model is then employed for score distillation. Specifically, the authors interpolate between the original and fine-tuned weights to generate intermediate score networks, ensuring a better alignment with the distribution of images rendered from partially completed 3D scenes.

**Strengths:**

+ This paper is well-written and provides strong motivating examples. The methodology is clearly presented and easy to follow, and the experiments section includes all necessary details on the experimental setup. The appendix is also comprehensive, offering background information and implementation details that enhance the paper's self-contained nature.

+ The proposed method is simple and straightforward to implement, comprising two standard stages: (1) fine-tuning the diffusion model with checkpointed weights and (2) performing interval score distillation using the diffusion model interpolated from these checkpoints.

**Weaknesses:**

- Sections 4.1 and 4.2 appear inconsistent. The final method in Section 4.2 seems to only save the last checkpoint and interpolate parameters to generate intermediate score networks. If this is the case, the path checkpointing scheme introduced in Sec. 4.1 and Alg. 1, where checkpoints are saved at regular intervals, may be unnecessary. Additionally, it would strengthen the paper to include an experiment comparing interpolated LoRA with path checkpointing at fixed intervals.

- The comparison with LucidDreamer seems not fair; currently, the proposed method benefits from initializing the 3D representation with a pre-trained generator. To ensure fairness, LucidDreamer should be evaluated using the same initialization.

- The ablation studies are limited. The choice of interval score distillation is not sufficiently justified, particularly given other possible distillation methods such as SDS, VSD, and related approaches [1, 2, 3].

[1] Katzir et al., Noise-free Score Distillation

[2] Wang et al., SteinDreamer: Variance Reduction for Text-to-3D Score Distillation via Stein Identity

[3] Wang et al., Taming Mode Collapse in Score Distillation for Text-to-3D Generation

- Although the paper claims faster convergence in both the abstract and introduction, there is no empirical evidence provided to support this claim.

**Questions:**

1. In Sec. 3, the STF is considered as a more accurate score function to highlight the OOD issue with the pre-trained score network. It would be insightful to understand how incorporating rendered images from the SF3D as references in STF impacts score distillation performance.

2. The motivation around addressing OOD appears to contradict prior studies [1, 2], which suggest that the pre-trained score function naturally brings OOD samples from 3D representations toward the in-domain distribution. If the diffusion model is sufficiently tuned on a well-initialized 3D structure, it may offer little effect in further improving the quality of rendered views.

3. It would also be beneficial to draw a connection to VSD, where the score network is dynamically fine-tuned to align with the distribution of rendered images. Clarifying how PTD fundamentally differs from VSD would strengthen the paper. Specifically, it would be useful to explain any distinctions in methodology that set PTD apart from VSD in managing distribution shifts during score distillation.

[1] Wang et al., Score Jacobian Chaining: Lifting Pretrained 2D Diffusion Models for 3D Generation

[2] Katzir et al., Noise-Free Score Distillation

---

### Note · Authors · 2024-12-04

I have read and agree with the venue's withdrawal policy on behalf of myself and my co-authors.